# Increased nitrogen enrichment and shifted patterns in the world's grassland: 1860-2016

Rongting Xu[1], Hanqin Tian[1,2], Shufen Pan[1], Shree R. S. Dangal[3,1], Jian Chen[4,1], Jinfeng Chang[5], Yonglong Lu[2], Ute Maria Skiba[6], Francesco N. Tubiello[7] and Bowen Zhang[1#]

[1]International Center for Climate and Global Change Research and School of Forestry and Wildlife Sciences, Auburn University, Auburn, AL 36849, USA

[2]Research Center for Eco-Environmental Sciences, State Key Laboratory of Urban and Regional Ecology, Chinese Academy of Sciences, Beijing 100085, China

[3]Woods Hole Research Center, Falmouth, Massachusetts 02540, USA

[4]Department of Computer Science and Software Engineering, Samuel Ginn College of Engineering, Auburn University, Auburn, AL 36849, USA

[5]Laboratoire des Sciences du Climat et de l'Environnement, LSCE, 91191 Gif sur Yvette, France

[6]Centre for Ecology and Hydrology, Bush Estate, Penicuik EH26 0QB, United Kingdom;

[7]Statistics Division, Food and Agricultural Organization of the United Nations, Via Terme di Caracalla, Rome 00153 Italy

[#]Present Address: Department of Natural Resources and Environmental Management, Ball State University, 2000 W. University Ave., Muncie, IN 47306, USA

*Correspondence to*: Dr. Hanqin Tian (tianhan@auburn.edu)

**Abstract**

Production and application to soils of manure excreta from livestock production significantly perturb the global nutrient balance and result in significant greenhouse gas emissions that warm the earth's climate. Despite much attention paid to synthetic nitrogen (N) fertilizer and manure N applications to croplands, spatially-explicit, continuous time-series datasets of manure and fertilizer N inputs on pastures and rangelands are lacking. We developed three global gridded datasets at a resolution of $0.5° \times 0.5°$ for the period 1860–2016 (i.e., annual manure N deposition (by grazing animals) rate, synthetic N fertilizer and N manure application rates), by combining annual and 5-arc minute spatial data on pastures and rangelands with country-level statistics on livestock manure, mineral and chemical fertilizers, and land use information for cropland and permanent meadows and pastures. Based on the new data products, we estimated that total N inputs, sum of manure N deposition, manure and fertilizer N application to pastures and rangelands increased globally from 15 to 101 Tg N yr$^{-1}$ during 1860–2016. In particular during the period 2000-2016, livestock manure N deposition accounted for 83% of the total N inputs, whereas manure and fertilizer N application accounted 9% and 8%, respectively. At the regional scale, hotspots of manure N deposition remained largely similar during the period 1860–2016 (i.e., southern Asia, Africa, and South America), however hotspots of manure and fertilizer N application shifted from Europe to southern Asia in the early 21st century. The new three global datasets contribute to fill previous data gaps of global and regional N inputs in pastures and rangelands, improving the abilities of ecosystem and earth system models to investigate the global impacts of N enrichment due to agriculture, in terms of associated greenhouse gas emissions and environmental sustainability issues. Datasets are available at https://doi.pangaea.de/10.1594/PANGAEA.892940.

24    **Keywords:** nitrogen, manure, fertilizers, pasture, rangeland, FAOSTAT

## 1 Introduction

Livestock production has increased substantially in response to growing meat consumption across the globe in the past century (Bouwman et al., 2013; Dangal et al., 2017). Agriculture occupies 37% of Earth's ice-free land surface for use as cropland and permanent meadows and pastures (Tubiello, 2018). Land used by livestock for permanent meadows and pastures is the largest component, using 25% of the total land earth surface (FAOSTAT, 2018) to generate 33%−50% of world total agricultural GDP (Herrero et al., 2013). While livestock is a major source of income for more than 1.3 billion people, it is also a major user of crop and freshwater resources (Dangal et al., 2017; Herrero et al., 2013). Overall, livestock production plays a major role as driver of global change in land use and nutrient cycles (Havlík et al., 2014; Herrero et al., 2013; Zhang et al., 2017). There is a growing recognition that livestock production is linked to increasing global greenhouse gas (GHGs) and ammonia emissions (Tian et al., 2016; Tubiello et al., 2018; Xu et al., 2018). Unsustainable practices, especially in intensive systems, may lead to severe pollution of aquatic systems and soil degradation locally, regional and globally, in particular through nitrate leaching to water bodies (Dangal et al., 2017; Davis et al., 2015; Fowler et al., 2013; Yang et al., 2016). Growing global demand for livestock products has increased grain production for feed in many regions, and has become a global driver of fertilizers trends, through increase in manure availability and synthetic fertilizer N use (FAOSTAT, 2018).

Livestock production systems therefore play an important role in global nutrient cycles. For example, nitrogen excretion from livestock increased from 21 Tg N yr$^{-1}$ in 1860 to 123 Tg N yr$^{-1}$ in 2016 (FAOSTAT, 2018; Zhang et al., 2017). Livestock contribute roughly two-thirds of non-$CO_2$ GHG emissions from agriculture (Smith et al., 2014), with roughly an equal share of $CH_4$ and $N_2O$ emissions (Dangal et al., 2017; Tubiello et al., 2013). Importantly, about 45% of total

anthropogenic $N_2O$ emissions are linked to manure deposited through grazing and manure
applied to croplands or left on pasture (Davidson, 2009; FAOSTAT, 2018). Globally, emissions
from manure N applied to soils or left on pastures has increased from 0.44 to 0.88 $GtCO_2eq\ yr^{-1}$
during 1961–2010 (FAOSTAT, 2018). Increased meat and dairy products consumption
worldwide was a major driver behind the documented increase in cattle herds globally
(FAOSTAT, 2018), and thus a major cause in the observed atmospheric increase of $N_2O$ and
$CH_4$ over the past several decades (Bai et al., 2018; Bouwman et al., 2013; Dangal et al., 2017;
Tubiello, 2018).
While the availability of national-level statistics is a fundamental component of our knowledge
base, environmental problems related to nitrogen pollution or emissions are best tackled at local
scale and often require finer, geo-spatial information, for example to assess proximity to water
bodies and thus pollution risks. In particular, a number of studies have focused on downscaling
existing national information to develop geospatially explicit regional and global datasets of
nitrogen fertilizer and livestock manure production and use, to better understand their feedback
on the climate system. Several datasets of N fertilizer use were used in this study, in particular
the FAOSTAT annual, country-specific statistics on mineral and chemical fertilizers and
livestock manure over the period 1961-2016 (FAOSTAT, 2018), as well as specific geospatially-
downscaled products (e.g., Bouwman et al., 2005; Lu and Tian, 2017; Mueller et al., 2012;
Nishina et al., 2017; Potter et al., 2010; Sheldrick et al., 2002). Further, global manure
production datasets were developed in different studies to achieve various research goals
(Bouwman et al., 2009; Bouwman et al., 2013; Holland et al.; Potter et al., 2010; Zhang et al.,
2017). Although datasets of manure application in croplands are increasingly available, there is
considerable uncertainty in the estimation of total manure application and their spatial
distribution across different studies (Gerber et al., 2016; Herrero et al., 2013; Liu et al., 2010;
Zhang et al., 2017).
Although previous studies have provided spatially explicit datasets of N inputs in the form of
mineral or chemical and manure N in cropland systems, the spatially explicit datasets on N
inputs in grassland systems are still missing (Hauglustaine, 2016; Lassaletta et al., 2014; Stehfest
and Bouwman, 2006). By grassland systems we mean the FAO livestock land use definition, i.e.,
land used as permanent meadows and pastures (FAOSTAT, 2018). The same may also be
referred to in the literature as 'pastures and rangelands'. We note that 'grassland' is in fact a land
cover definition. In order to avoid the confusion often made in the literature between land cover
and land use terminology, we will adopt FAO land use terminology of 'permanent meadows and
pastures,' to which the various national regional and global land use statistics cited in this work
refer. Furthermore, using results from the HYDE 3.2 dataset (Klein Goldewijk, 2017), we may
split the FAO land use category into 'pastures' and 'rangelands', to highlight differences
between managed intensive and unmanaged extensive systems, as needed. To enhance our
understanding of the role of livestock on the global GHG balance and nutrient budgets (e.g.,
ammonia emissions, nitrate leaching), global biogeochemistry models require spatially explicit
estimates of N inputs. In this study, we developed datasets for major sources of N inputs in
agriculture (i.e., manure and fertilizer application and manure deposition on permanent meadows
and pastures), using the recently published FAOSTAT statistics on manure N use in agriculture
(FAOSTAT, 2018). The latter are estimates based on IPCC Tier 1 methodology, i.e., they rely on
default coefficients prescribing, among other variables, N excretion rates by animal type and
region, as well as regional compositions of manure management systems (FAOSTAT, 2018).
Through combining the land-use dataset HYDE 3.2, FAOSTAT fertilizers N statistics, and
gridded manure production data in Zhang et al. (2017), we developed three annual global
datasets at a spatial resolution of $0.5° \times 0.5°$, as follows: 1) manure N application rates to
pastures (1860–2016); 2) synthetic N fertilizer application rates to pastures (1961–2016); and 3)
manure deposition rates by grazing livestock, to rangelands and pastures (1860–2016). We
quantified regional variations in N inputs, identified hotspots of N inputs from different N
sources from livestock, and discussed their uncertainty. These datasets are developed for global
model simulation studies in model inter-comparison projects (e.g., NMIP; Tian et al., 2018a;
Tian et al., 2018b), and will be updated annually based on regular annual updates of FAO
fertilizers and land use statistics and other sources of data such as global land use data products.
**2 Methods**
2.1 Land Use Categories
The concepts of grassland, pastures and meadows span several international land cover and land
use statistical definitions, specifically those used by FAO (FAOSTAT, 2018). In this paper, we
follow the relevant FAO land use definition of 'permanent meadows and pastures,' considering
our focus on livestock production. Importantly, complete country, regional and global statistics
available from FAO refer to this land use category. This land use definition is roughly equivalent
to the one adopted by the academic community engaged in global biogeochemical modeling, for
which 'grassland systems' are thought of as land cover/land use areas dominated by herbaceous
and shrub vegetation, including savannas (Africa, South America and India), steppes (Eurasia),
prairies (North America), shrub-dominated areas (Africa), meadows and pastures (United
Kingdom and Ireland) and tundra (Breymeyer, 1990; White et al., 2000).
For mineral and chemical fertilizers, we further split the FAO definition using HYDE 3.2, into
'pastures' and 'rangelands,' the former representing land use areas managed to support high
stocking densities of grass production for hay/silage, whereas the latter represents unmanaged
and grazed at low stocking densities. Although FAOSTAT land-use statics cover in principle
these two sub-categories of land use, data coverage needed is insufficient for the consistent
global mapping needed herein. The spatial distribution map of pastures and rangelands provided
by HYDE are nonetheless based on and normalized to FAOSTAT land use statistics,
complemented by additional information (Klein Goldewijk, 2017). To investigate N inputs from
livestock at a regional level, the global landmass was disaggregated into seven regions: North
America, South America, Africa, Europe, southern Asia (i.e., west, south, east, central and
southeast Asia), northern Asia, and Oceania (Fig. S1).
## 2.2 Global synthetic fertilizer N application on pastures
We obtained national-level datasets of "Agricultural use of mineral or chemical fertilizers" from
the FAOSTAT (2018) 'Fertilizers by Nutrient' domain, over the time series 1961−2016. The
FAOSTAT statistics of agricultural use include use for both agriculture and forestry, as well as
use in aquaculture. Furthermore, agricultural use includes both cropland and permanent
meadows and pastures. We assumed that fertilizers use for forestry and aquaculture was zero, as
well as fertilizers applications on rangelands. Subsequently, we estimated N application rates to
pastures by using the ratio of pasture to cropland N use total published by Lassaletta et al. (2014).
We finally spatialized the pasture N data using HYDE 3.2, obtaining gridded maps of synthetic
fertilizers N application rates on pastures in each grid cell area, over the period 1961−2016 (Fig.
1). We assumed even application rates within each country. Although gridded livestock density
maps were available from FAO, these are currently fixed for specific time periods, mainly 2010,
so that we deemed their use not particularly relevant to improve estimates for the 1961-2016
time series considered herein. Improved live density map products from FAO will considerably
improve our work and reduce uncertainty, and will be used when available.
## 2.3 Global manure N application to pastures
We obtained country-level datasets of "manure applied to soils" from the FAOSTAT (2018)
'Livestock Manure' domain for the period 1961−2016 (FAO, 2018). Following IPCC guidelines,
the data in this domain do not consider N leaching during treatment (FAOSTAT, 2018).
Furthermore, the FAOSTAT data do not separate manure application to cropland and pastures
and data of manure N application rates to pastures are currently not available. We therefore
assumed that manure N application rates in pastures and croplands were the same, considering
that the overall uncertainty in the input manure N data would not justify further assumptions at
this stage of knowledge. Improved FAO statistics on both use and application rates will be used
when available to improve this current work. Through combining land-use data HYDE 3.2, we
calculated the total cropland and pasture areas within each country where manure application
amount was larger than zero. We then computed mean manure N application rates on pastures,
annually over the period 1961−2016 (Fig. 2).
We calculated the national-level ratio of manure application to production ( $R_{a2p_{y,j}}$ ) by
combining gridded manure production data in Zhang et al. (2017) and the grid cell area. To
spatialize the national-level manure N application amounts to gridded maps of application rates
in each grid area, we multiplied the $R_{a2p_{y,j}}$ in grids where pasture areas were larger than zero
with the time-series gridded spatial distribution maps of manure production rate in Zhang et al.
(2017) during 1961–2014 and based on the spatial distributions of global pastures in land-use
data HYDE 3.2 (Klein Goldewijk et al., 2017).
The above-mentioned processes are represented by following equations:
$$R_{a2p_{y,j}} = \frac{T_{Mapp_{y,j}}}{\sum_{g=1}^{g=n\ in\ country\ j}(R_{Mprod_{y,g}} \times A_g)}$$
(1)

where year is from 1961 to 2016, and country number is 165. $R_{a2p_{y,j}}$ is the ratio (unitless) of
manure application to production in the year $y$ and country $j$. $T_{Mapp_{y,j}}$ is the national total manure
application amount (kg N yr$^{-1}$) derived from the FAO database for each year. $A_g$ is the area of
each grid (km$^2$).
$$R_{Mapp_{y,g}} = R_{a2p_{y,j}} \times R_{Mprod_{y,g}}$$
(2)

where $R_{Mapp_{y,g}}$ is the gridded manure application rate (kg N km$^{-2}$ yr$^{-1}$) in year $y$ and country $j$.
As the national-level manure application amount was not available during 1860–1960, we
assumed that $R_{a2p_{y,j}}$ is the same as for 1961. Combining with the gridded spatial maps of
manure production rates in Zhang et al. (2017), we generated the datasets of spatialized manure
application rates to global pastures during 1860–1960.
Finally, we calculated manure application amounts in each country by combining $R_{Mapp_{y,g}}$ and
grid areas to compare with national-level deposition amounts from the FAOSTAT database
during 1961–2016. As we calculated national-level manure application amounts during
1860–1960 using $R_{a2p_{y,j}}$ in 1961, these data served as national total manure N application
amounts to adjust $R_{Mapp_{y,g}}$ during 1860–1960.
The adjustment procedure is represented in the following equations:

$$CT_{Mapp_{y,j}} = \sum_{g=1}^{g=n\ in\ country\ j}(R_{Mapp_{y,g}} \times A_g) \qquad (3)$$

where year is from 1860–2016. $CT_{Mapp_{y,j}}$ (kg N yr$^{-1}$) is the calculated national-level manure

application amounts in the year $y$ and country $j$. If $CT_{Mapp_{y,j}}$ is less or more than $T_{Mapp_{y,j}}$, an

adjustment is needed to keep calculated national total amounts consistent with amounts from the

FAOSTAT database. In this case, $CT_{Mapp_{y,j}}$ is less than $T_{Mapp_{y,j}}$ using Eq. 3, thus an adjustment

is needed, using the following equations:

$$R_{a_{y,j}} = \frac{T_{Mapp_{y,j}}}{CT_{Mapp_{y,j}}} \qquad (4)$$

where $R_{a_{y,j}}$ is the regulation ratio (unitless) in the year $y$ and country $j$.

$$R_{Mapp_{y,g\ (r)}} = R_{Mapp_{y,g}} \times R_{a_{y,j}} \qquad (5)$$

where $R_{Mapp_{y,g\ (r)}}$ is real gridded manure application rate (kg N km$^{-2}$ yr$^{-1}$) in the year $y$ and

country $j$.

## 2.4 Global manure N deposition on pastures and rangelands

To develop global distribution maps of manure N deposition by grazing animals, we first

obtained country-level statistics of "manure left on pasture" over the period 1961–2016 from the

FAOSTAT (2018) 'Livestock manure' domain of FAOSTAT agri-environmental indicators

(FAO, 2018). We then obtained the national-level ratio of manure deposition to production

($R_{d2p_{y,j}}$) by combining country-level FAOSTAT datasets of "Manure left on pasture" and

gridded total manure production datasets based on Zhang et al. (2017). Then, we used spatial

distributions of global permanent meadows and pastures, including pastures and rangelands,

based on HYDE 3.2 grassland data (Klein Goldewijk, 2017) and gridded maps of deposition

rates, to spatialize the national-level manure N deposition at the global scale. For example, we
multiplied the $R_{d2p_{y,j}}$ ratio in grids within which the pastures and rangelands area was larger
than zero, with the time-series gridded spatial distribution maps of manure production rates in
Zhang et al. (2017) during 1961–2014 (Fig. 2).
The above-mentioned processes are represented by the following equations:
$$R_{d2p_{y,j}} = \frac{T_{Mdep_{y,j}}}{\sum_{g=1}^{g=n\ in\ country\ j}(R_{Mprod_{y,g}} \times A_g)} \tag{6}$$

where year ($y$) is from 1961 to 2016 and country number ($j$) is 157. $R_{d2p_{y,j}}$ is the ratio (unitless)
of manure deposition to production in the year $y$ and country $j$. $T_{Mdep_{y,j}}$ is national total manure
deposition amount (kg N yr$^{-1}$) derived from the FAOSTAT database for each year. $R_{Mprod_{y,g}}$ is
the gridded manure N production rate (kg N km$^{-2}$ yr$^{-1}$) in the year $y$ and grid $g$.
$$R_{Mdep_{y,g}} = R_{d2p_{y,j}} \times R_{Mprod_{y,g}} \tag{7}$$

where $R_{Mdep_{y,g}}$ is the gridded manure deposition rate (kg N km$^{-2}$ yr$^{-1}$) in the year $y$ and country $j$.
Finally, we calculated the manure deposition amount for each country through combining
$R_{Mdep_{y,g}}$ and grid area to compare with the national-level deposition amounts from the
FAOSTAT database, using the following equation:
$$CT_{Mdep_{y,j}} = \sum_{g=1}^{g=n\ in\ country\ j}(R_{Mdep_{y,g}} \times A_g) \tag{8}$$

where $CT_{Mdep_{y,j}}$ (kg N yr$^{-1}$) is the calculated national-level manure deposition amount in the
year $y$ and country $j$. If $CT_{Mdep_{y,j}}$ is less or more than $T_{Mdep_{y,j}}$, an adjustment was made to keep
calculated national total amounts consistent with those from the FAOSTAT database. In this case,
$CT_{Mdep_{y,j}}$ is roughly equal to $T_{Mdep_{y,j}}$ using Eq. 8, thus no adjustment was needed.
Since the national-level manure deposition amounts are not available during 1860−1960, we
assumed that $R_{d2p_{y,j}}$ is the same as that in 1961. Combining the gridded spatial maps of manure
production rates in Zhang et al. (2017), we generated datasets of spatialized manure deposition
rates on permanent meadows and pastures globally, for the period 1860−1960.

## 3 Results

## 3.1 Synthetic fertilizer N application to pastures, 1961−2016

The FAO data, combined with the geospatial analysis in this work, show that the total amount of
synthetic N fertilizer applied to pastures increased from 0.04 to 8.7 Tg N yr$^{-1}$ during 1961−2016
at an average rate of ~0.18 Tg N yr$^{-1}$ ($R^2$ = 0.98) per year (Fig. 3a). Synthetic N fertilizer
application rates showed rapid increases across the globe, with large spatial variations during the
study period (Figs. 4b-c). The global average application rate on pastures was 0.07 kg N ha$^{-1}$ yr$^{-1}$
in 1961 and reached 10.9 kg N ha$^{-1}$ yr$^{-1}$ in 2016 (increased ~154 folds) (Table 1).
In the 1960s, Europe (0.2 Tg N yr$^{-1}$) was the largest contributor (67.8%) to the total global N
fertilizer use, followed by North America (0.06 Tg N yr$^{-1}$, 21.8%) and southern Asia (0.03 Tg N
yr$^{-1}$, 9.9%) (Fig. 5a). The remaining regions accounted for less than 1% of the total N fertilizer
application. During 1961−2016, southern Asia showed a continuous increase of N fertilizer
consumption and became the largest contributor (3.4 Tg N yr$^{-1}$, 45%) between 2000 and 2016.
In contrast, Europe's synthetic N fertilizer use and contribution to the global total decreased
since the 1980s (Fig. 5a). This is a well-known trend, linked to EU-wide policy directives aimed
at minimizing N pollution (Tubiello, 2018). During 2000−2016, Europe applied 2.1 Tg N yr$^{-1}$,
which accounted for 27% of the total global N fertilizer use on pastures. There was a slight
increase in the contribution from North America, and the synthetic fertilizer N use amount
increased by 1.6 Tg N yr$^{-1}$. The remaining regions accounted for roughly 7% of the total N
fertilizer application on pastures.
The average synthetic N application rate in Oceania, North America, and southern Asia showed a
rapid increase over the period 1961−2016 (Fig. 5d). Africa and northern Asia showed a slight
increase in average N fertilizer application rates during the study period.  Europe exhibited a
rapid increase of N fertilizer application rates since 1961, then decreased after 2000, and then
started to increase in recent five years (Fig. S3).
We identified the top five countries (India, United States, China, France, and Germany) with
highest fertilizer N application to pastures in 2016. These countries consumed 49% to 58% of the
total N fertilizer from 1961 to 2016. India (1.5 Tg N yr$^{-1}$) and the United States (1.5 Tg N yr$^{-1}$)
were the two largest contributors in 2016, at an increasing rate of 45 Gg N yr$^{-1}$ ($R^2$ = 0.98) per
year during 1980−2016 and 32 Gg N yr$^{-1}$ ($R^2$ = 0.99) per year during 1961−2016, respectively.
China consumed 1.4 Tg N yr$^{-1}$ in 2016 at an increasing rate of 34 Gg N yr$^{-1}$ ($R^2$ = 0.96) per year
during 1977−2016 while there was only a slight increase during 1961−1976. In contrast,
fertilizer N use in France peaked in 1999 (0.8 Tg N yr$^{-1}$), then showed a rapid decrease until
2016 (0.5 Tg N yr$^{-1}$). Similarly, in Germany, it peaked in 1988 (0.8 Tg N yr$^{-1}$), and showed a
continuous decrease until 2016 (0.3 Tg N yr$^{-1}$).
3.2 Manure N application to pastures, 1860−2016
Our results showed that the annual manure N application rates on pastures increased from 1.4 to
8.6 Tg N yr$^{-1}$ during 1860−2016 (Fig. 3a). Manure N application rates showed rapid increases
across the globe and exhibited large spatial variations, shifting the regional use from North
America and Europe to Asia, during the study period (Figs. 4d-f). The global average manure
application rate was 5.3 kg N ha$^{-1}$ yr$^{-1}$ in the 1860s and roughly doubled by 2016 (10.7 kg N ha$^{-1}$
yr$^{-1}$) (Table 1).
From the regional perspective (Fig. 5b), in the 1860s Europe (0.8 Tg N yr$^{-1}$) was the largest
contributor and accounted for 53%, while southern Asia (0.25 Tg N yr$^{-1}$) accounted for 17% of
the global total manure N application on pastures. South and North America shared the same
proportion (13%), whereas the remaining regions only shared 4%. Conversely during 2000−2016,
manure N application on pastures in southern Asia (2.9 Tg N yr$^{-1}$) was tenfold higher than that in
the 1860s and accounted for 36% of the global total, surpassing Europe, which accounted for 28%
of the global total. Manure N application amounts in North America and South America
increased, but with different magnitudes. During 2000−2016, North America accounted for 11%,
while South America accounted for 17% of the global total. In the remaining regions, significant
increases of annual manure N application on pastures also occurred, but their contributions to the
global total changed only slightly (8%) compared to the 1860s.
The regional average manure N application rate was increasing in southern Asia and Africa
during 1860−2016 (Fig. S3b). South America, Oceania, and North America exhibited a rapid
decreasing trend of manure N application rates from the 1860s to the 1960s and showed
continuous increases afterward until 2016 (Figs. 5e, S3b), which was associated with the
substantial expansion of pasture areas (Table S2). Europe exhibited a rapid increase of manure N
application rates since the 1860s, then decreased after the 1980s (Figs. 5e).
In 2016, the top five countries with largest manure N applications on pastures were China,
United States, Brazil, Russia, and France. Manure N application in these countries contributed 43%
to 52% of global total use from 1961 to 2016. China (2.5 Tg N yr$^{-1}$) alone accounted for 30% in
2016 at an increasing rate of 42 Gg N yr$^{-1}$ ($R^2$ = 0.98) per year during 1961−2016. Manure N use
on pastures in Brazil and the United States was roughly the same (0.7 Tg N yr$^{-1}$) in 2016. Both
countries showed a slower increasing trend (Brazil: 7 Gg N yr$^{-1}$ per year and United States: 3 Gg
N yr$^{-1}$ per year) during 1961–2016. In contrast, Russian manure N application peaked in 1989
(0.7 Tg N yr$^{-1}$), then showed a rapid decrease until 2016 (0.3 Tg N yr$^{-1}$). Similarly, in France, it
peaked in 1979 (0.45 Tg N yr$^{-1}$), then showed a continuous decrease until 2016 (0.28 Tg N yr$^{-1}$).
3.3 Manure N deposition on pastures and rangelands, 1860–2016
Our data show that the total amounts of manure N deposited on pastures and rangelands
increased from 14 to 84 Tg N yr$^{-1}$ during 1860–2016 (Fig. 3b). Manure N deposition rates
increased steeply across the globe, but exhibited large spatial variations during the study period
(Fig. 4g-i). The increase was much larger in the eastern world (typically China and India) and
South America compared to the western world. The global average manure deposition rate was
11 kg N ha$^{-1}$ yr$^{-1}$ in 1860 and reached 25 kg N ha$^{-1}$ yr$^{-1}$ in 2016 (Table 1).
At the regional scale (Fig. 5c), in the 1860s southern Asia was the region with the largest manure
N deposition on pastures and rangelands (4.4 Tg N yr$^{-1}$; 30% of total manure N deposition
amounts), followed by Africa (2.8 Tg N yr$^{-1}$; 19%) and South America (2.4 Tg N yr$^{-1}$; 16%).
Manure N deposition in the remaining regions was estimated to be 5.1 Tg N yr$^{-1}$, contributing 35%
to the total manure N deposition amount. During 2000–2016, southern Asia, Africa, and South
America were still the three largest contributors: 27 Tg N yr$^{-1}$ accounted for 34%, 20 Tg N yr$^{-1}$
accounted for 26%, and 15 Tg N yr$^{-1}$ accounted for 20% of the global manure N deposition on
pastures and rangelands, respectively. The remaining regions (Oceania, North America, and
Europe) contributed to 20% of the global total during 2000–2016. Europe and Oceania saw an
increase in manure N deposition amounts from 1860 to 1960, but since 1980 there was a
significant decrease, partly explained by the onset of N pollution regulation. Manure N
deposition amounts in North America increased during 1860−1980, but changed slightly since

310    1960.

Oceania showed a continuously decreasing trend of average manure N deposition rates in
pastures and rangelands over the period 1860−2016. Manure N deposition rates in South
America decreased between 1860 and 1960 and then increased afterward until 2016 (Fig. S3c).
The significant contrast of changes in manure N deposition rates in Oceania and South America
between the 1860s and the 1960s is due to the substantial and rapid increase of grassland areas
(Tables S2, S3). Africa and southern Asia saw  continuous increases in manure N deposition
rates from 1860 to 2016, whereas Europe and North America was found with decreasing
deposition rates since the 1980s (Figs. 5f, S3c).
In this study, we identified the top 10 countries (China, Brazil, India, Ethiopia PDR, United
States, Australia, Sudan (former), Pakistan, Argentina, and Nigeria) that together contributed to
48% of the global total manure N deposition on pastures and rangelands in 2016. Among these
countries, China (17%) and Brazil (21%) were the two largest contributors, with the similar
annual rate of increase of ~125 Gg N yr$^{-1}$ (R$^2$ = 0.99) per year during 1961−2016. India was the
third largest contributor, however, at a small increasing rate of 63 Gg N yr$^{-1}$ (R$^2$ = 0.98) per year
during 1961−2016. Annual manure N deposition in Ethiopia PDR was stable during 1961−2000,
but since then rapidly increased at a rate of 117 Gg N yr$^{-1}$ (R$^2$ = 0.96) per year. The United States
showed a significant increase of annual manure N deposition on pastures and rangelands from
1961 to 1975 and then was stable after 1980. Australia showed a decreasing trend during
1990−2016 at a rate of 62 Gg N yr$^{-1}$ (R$^2$ = 0.92) per year, whereas, in the former Sudan, Pakistan,
and Nigeria annual manure N deposition amounts to pastures and rangelands increased at an
annual average rate of 68 ($R^2$ = 0.8), 46 ($R^2$ = 0.97), 56 ($R^2$ = 0.98) Gg N yr$^{-1}$ per year,
respectively. There was no significant change in manure N deposition amounts in Argentina; the
annual from 1961 to 2016 was 2.6 Tg N yr$^{-1}$.

**4 Discussion**

4.1 Overview of global N inputs to pastures and rangelands

The global N cycle has been significantly perturbed by human activity since at least the
industrial revolution. Intense agricultural activities, such as synthetic N fertilizer production and
use, and intensive livestock production, were identified as major drivers to such change. In this
context, improving estimates of global anthropogenic N inputs to pastures and rangelands and
their consequences, including on $N_2O$ emissions, is important (Galloway et al., 2008; Tian et al.,
2016; Xu et al., 2017). In this study, we generated global datasets of fertilizers N inputs from
livestock, both synthetic and from manure, during the period 1860−2016. Pastures and
rangelands have experienced substantial land expansion over the period of 1860−1998 (Klein
Goldewijk, 2017). The total amount of mineral and manure N applied to permanent meadows
and pastures increased by 573% over the study period, from 15 to 101 Tg N yr$^{-1}$ from 1860 to
2016. During 2000−2016, the global mineral N fertilizer application to agriculture was
significant, reaching 110 Tg N yr$^{-1}$ in 2016, while manure N production was 123 Tg N yr$^{-1}$ (FAO,
2018; FAOSTAT, 2018), resulting in a total input of 233 Tg N yr$^{-1}$. Our estimate of total N
inputs (synthetic N fertilizer: 7.5 Tg N yr$^{-1}$; manure N application: 8.2 Tg N yr$^{-1}$; manure N
deposition: 78.1 Tg N yr$^{-1}$) to permanent meadows and pastures (93.8 Tg N yr$^{-1}$) accounted for
45% of global total N production (manure: 114.2 Tg N yr$^{-1}$; synthetic N fertilizer: 96.4 Tg N yr$^{-1}$)
during 2000−2016.

## 4.2 Extension of FAO information

Our work extends the relevant FAO national-level statistics in order to provide input drivers for process-based model simulations (e.g., $N_2O$-MIP, Tian et al., 2018a; Tian et al., 2018b). We furthermore separated N application rates between pastures and cropland, based on previous published work. We likewise extended information available in FAOSTAT by providing spatialized manure N application rates to pastures and spatialized national-level manure N deposition dataset from 1860 to 2016.

## 4.3 Comparison with other studies

We compared our datasets with other existing data sources (Table 2). Our estimate of world total manure N use on pastures was 58% and 171% higher than that estimated by Stehfest & Bouwman (2006) and Liu et al. (2010), respectively. However, our estimate was 39% and 87% lower than estimates by Bouwman et al. (2002 and 2013, respectively). Critically, pasture area data varied significantly across different studies. For example, Bowman et al. (2013) divided grasslands into mixed and pastoral systems, and estimated grasslands area based on the country- or regional-level grazing intensity (Table 2). In addition, synthetic fertilizers were applied to the area of mixed agricultural systems (grassland and cropland) and manure N was assumed to be applied to both mixed and pastoral systems. The HYDE 3.2 land use dataset divides the global grazing area into intensively managed grasslands (pastures), and less intensive and unmanaged grasslands (rangelands) (Klein Goldewijk et al., 2017). In this study, we rather assumed that all manure N was applied to pastures, the latter estimated from the HYDE database (798 Mha). Hence, pasture area defined in Bowman et al. (2013) was more than fourfold higher than the data

we used. Consequently, the spatial distribution and annual total N application differed
substantially compared with that in Bowman et al. (2013).
Similarly, the estimates of N fertilizer use in pastures showed large variations across studies
(Table 1). This study obtained country-level N fertilizer amounts applied to pastures from the
national-level ratios provided by Lassaletta et al. (2014) and total N amounts applied to soils
provided by FAOSTAT. Thus, the global N fertilizer amount in 2000 was consistent with that in
Lassaletta et al. (2014). Liu et al. (2010) assumed that 16% of fertilizer was applied to global
grasslands. Their estimate was roughly twice as high as this study (6.2 Tg N yr$^{-1}$) for the year
2000. The estimates by Bowman et al. (2002) and Stehfest & Bouwman (2006) were 31% and
50%, respectively, lower than our estimates in the corresponding years. Klein Goldewijk et al.
(2017) divided land used for grazing into more intensively used pastures, less intensively used or
unmanaged rangelands. In this study, we assumed N fertilizer was applied to all global pastures
and therefore the total area of intensively managed grassland was significantly different from the
area used in Bowman et al. (2002) and Chang et al. (2016).
## 4.4 Changes in N inputs hotspots
Overall, southern Asia ranks as a top hotspot of all sources of global N inputs in pastures and
rangelands during the past three decades, causing a major threat to environmental sustainability
and human health in this region. In the 1860's overall manure N production amounts were
similar in Asia and Europe (Zhang et al., 2017). However, manure N deposition was 2.4 times
higher than that in Europe, whereas manure N application was roughly three times lower than
that in Europe. During 2000−2016, southern Asia accounted for ~42% of global manure N
production. Consequently, manure N deposition and application amounts in southern Asia were
the highest compared to the rest of the regions between 2000−2016. These increases are due to
large increases in animal numbers (e.g., cattle, sheep and goats) since 1950 (Bouwman et al.,
2013; Dangal et al., 2017). For the rest of the regions, the increases of livestock numbers were
also found in South America and Africa since 1860, whereas livestock numbers in Europe and
North America showed a decreasing trend after 1980 (Dangal et al., 2017). Thus, besides
southern Asia, South America and Africa were hotspots for manure N deposition during
1860–2016, while manure N deposition amount decreased in Europe and North America since
the 1980s.
4.4.1 Shifting hotspots of N fertilizer application
European countries (e.g., Germany, United Kingdom, and Ireland) were identified as top
hotspots of global N fertilizer application in 1961 (Fig. 4b). However, these hotspots have shifted
from Western Europe towards southern Asia at the end of the 20th century (Fig. 4c). Southern
Asia was found with the highest N fertilizer application amounts between 2000 and 2016, most
concentrated in countries of East and South Asia (e.g., China and India). China and India
together applied 36% of global total N fertilizer to pastures and rangelands.
4.4.2 Shifting hotspots of manure N application
Manure application hotspots moved from European countries to southern Asia during the past
155 years. Between 1860 and 1999, Europe accounted for 50% of global total manure N
application to pastures and experienced a rapid growth of manure N application, peaked (3.5 Tg
N yr$^{-1}$) in 1986. In 1860, the highest applications were in the United Kingdom, France, and
Germany (Fig. 4d), but by 2016, the highest application was in the North China Plain (Fig. 4f).
China alone applied 29% of global total manure N during 2000–2016.
4.4.3 Shifted hotspots in manure N deposition
Southern Asia, as the hotspot of manure N deposition to pastures and rangelands, contributed 31%
of the global total amount during the past 157 years. Also, in Africa and South America
substantial increases of manure N deposition during 1860−2016 were observed. In the 1860s,
manure N deposition from southern Asia, Africa, and South America contributed to 65%,
whereas Europe accounted only for 12% of the global total manure N deposition. In 1860, the
highest deposition rates were observed for New Zealand, Australia, and Western Europe (Fig.
4g). In 2016, except for the above-mentioned regions, the highest deposition rates were in South
and West Asia, China, West and East Africa, and South America (Fig. 4i). During 2000−2016,
manure N deposition from southern Asia, Africa, and South America contributed to 80%, while
Europe accounted for 5% of the global total amount.
## 4.5 Limitations and uncertainties
This study attempts to provide an overall estimate of N inputs to global rangelands and pastures,
during the period 1860−2016. However, before these data are used in global models,
uncertainties of these datasets need to be addressed. First, the different definitions of grassland
systems used by the scientific community introduce uncertainties of the spatial patterns and
annual total amounts of N inputs. Chang et al. (2016) generated global maps of grassland
management intensity since 1901 based on modeled net primary production and the use of grass
biomass generated by Herrero et al. (2013). Their total grassland area substantially differed from
pasture area developed by HYDE 3.1 (Chang et al., 2016). In this study, we used HYDE 3.2 to
generate N inputs to global grasslands, defined more appropriately by using the FAO land use
definition of 'permanent meadows and pastures'. This dataset exactly followed the FAOSTAT
data during 1960−2015, and combined population density data to reconstruct land use prior to
1960. Pastures and rangelands defined in HYDE 3.2 were based on the intensity of human
management. Although Bouwman et al. (2013) indicated that grassland areas in their study were
also calculated based on the grazing intensity, their total area (pastures and rangelands) and
spatial patterns were obviously different from HYDE 3.2 (Table 2). Thus, a better understanding
of land use is vital to reduce the uncertainty of estimating N input rates and amounts in pastures
and rangelands.
Second, the FAOSTAT database provides country-level manure N applied to soils; however, this
dataset could not be directly applied to study N cycles on pastures since applications to cropland
and pasture soils are not differentiated. In this study, it remains large uncertainty that we
separated national-level manure N application on pastures simply based on pasture area over
total agricultural area (cropland, pastures and rangelands). In previous studies, Bouwman et al.
(2013) assumed that 50% and only 5% of the available manure was applied to grasslands in most
industrialized countries and in most developing countries, respectively. Liu et al. (2010)
allocated 34% of the national total solid manure to pastures in European countries and Canada,
13% of the national total manure to pastures in the United States, and 10% of the national total
manure to pastures in developing countries. Chang et al. (2016) assumed that manure N
application rate changes along with changes in the total ruminant stocking density. Moreover, the
spatialization process of N application rates might introduce large uncertainty. The spatial
pattern of gridded manure N application rates in our study are correlated with manure production
rates in Zhang et al. (2017). The assumptions and uncertainties mentioned in their study, such as
without considering livestock migration, might cause uncertainty of spatial distribution.
Third, studies used different data sources and made various assumptions of the annual amount of
fertilizer N applied on global pastures (Bouwman et al., 2002; Chang et al., 2016; Lassaletta et
al., 2014; Liu et al., 2009; Stehfest and Bouwman, 2006). Thus, there remains large uncertainty
of total N application on permanent meadows and pastures globally. Moreover, N fertilizer
application rates by crops were highly investigated and documented in previous studies. Hence,
N fertilizer application datasets were generated considering crop-specific fertilizer rates and
cropland area in each grid (Lu and Tian, 2017; Mueller et al., 2012; Nishina et al., 2017; Potter et
al., 2010). In reality, N fertilizer application on pastures of each country is not homogeneous. In
this study, we assumed that N fertilizer application rate in each country was constant, which
means fertilizer was applied evenly in each grid with pastures area larger than zero. Last, inside
each relevant land use cell pastures and rangelands may be characterized by different livestock
density and deposition rates, which is not considered in our current datasets. The final manure N
deposition would be highly affected by the proportion of each type of management in the cell.
Thus, it is necessary to consider these in the future research.
Furthermore, other human-induced sources of N inputs to pastures and rangelands were not
included in our study, which may underestimate total N received globally. For example,
biological N fixation was one of the major N sources in the terrestrial ecosystem in the absence
of human influence (Cleveland et al., 1999). Pastures and rangelands occupy 25% of the Earth's
ice-free land surface across different latitudes with divergent biological N fixation abilities. Plant
production in temperate grasslands is proximately limited by N supply due to little N via N
fixation; however, tropical savannah received a large amount of N through leguminous species
(Cleveland et al., 1999; Vitousek et al., 2013). An estimate of potential N fixation amount by
global grassland systems is ~46.5 Tg N yr$^{-1}$, with a range of 26.6–66.5 Tg N yr$^{-1}$ (Cleveland et al.,
1999). Atmospheric N deposition is another major source of N input to permanent meadows and
pastures globally and increased from 2 to 14 Tg N yr$^{-1}$ for the period 1860–2016 based on the
Chemistry–Climate Model Initiative N deposition fields (Eyring et al., 2013; Tian et al., 2018a;
Tian et al., 2018b).

## 5 Data availability

The $0.5° \times 0.5°$ gridded global datasets of manure nitrogen deposition, manure nitrogen
application, and nitrogen fertilizer application in grassland systems are available at
https://doi.pangaea.de/10.1594/PANGAEA.892940 (Xu et al., 2018). Data are in ASCII format.
A supplemental file is added to the list of all other parameters used in this study to calculate
these three datasets in global grassland systems.

## 6 Conclusion

In the context of increasing livestock production, manure and fertilizer N inputs to permanent
meadows and pastures (pastures and rangelands areas) globally have increased rapidly since the
industrial revolution. However, datasets of global N inputs are still incomplete. This is the first
study that has attempted to consider major sources of anthropogenic N inputs in permanent
meadows and pastures and hence generated time-series gridded datasets of manure and fertilizer
N application rates, and manure deposition rate during 1860–2016. Our datasets indicated a rapid
increase of total N inputs to pastures and rangelands globally during this period, especially the
past half century. The hotspots of grassland N application shifted from European countries to
southern Asia, specifically China and India during 1860–2016, which indicated the spatial
transformation of environmental problems. In this study, we have obtained N data from various
sources to fill the data gap; however, large uncertainties still remain in our datasets (e.g., N
application rate within each country, annual manure application amounts). More information is
needed to improve these datasets in our further work.
**Acknowledgements:** This study has been supported by National Key R & D Program of China
(Grant Number: 2017YFA0604702, 2018YFA0606001), NOAA Grants (NA16NOS4780207,
NA16NOS4780204), National Science Foundation (1210360, 1243232), STS Program of the
Chinese Academy of Sciences (KFJ-STS-ZDTP-010-05), SKLURE Grant (SKLURE2017-1-6).
We are grateful to FAO and its member countries for the collection, analysis and dissemination
of fertilizers and land use statistics. We thank Dr. Wilfried Winiwarter in International Institute
for Applied Systems Analysis for constructive comments that have helped improve this study.

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

**Table 1.** The N input rates, applied/deposited area, and total amounts in global pastures and rangelands in
1860, 1961, 2000, and 2016 (1 $km^2$ = 100 ha).

| | 1860 | 1961 | 1980 | 2000 | 2016 |
|---|---|---|---|---|---|
| **Averaged N fertilizer application rate (kg N ha$^{-1}$ yr$^{-1}$)** | NA | 0.07 | 3.6 | 7.8 | 10.9 |
| **Total applied area (Mha)** | NA | 623.8 | 725 | 797.8 | 803.1 |
| **Total amounts (Tg N yr$^{-1}$)** | NA | 0.04 | 2.6 | 6.2 | 8.7 |
| **Average manure N application rate (kg N ha$^{-1}$ yr$^{-1}$)** | 5.3 | 8.1 | 9.8 | 9.5 | 10.7 |
| **Total applied area (Mha)** | 268.2 | 623.8 | 725 | 797.8 | 803.1 |
| **Total amounts (Tg N yr$^{-1}$)** | 1.4 | 5.0 | 7.1 | 7.6 | 8.6 |
| **Average manure N deposition rate (kg N ha$^{-1}$ yr$^{-1}$)** | 11.2 | 15.4 | 19.0 | 20.7 | 25.3 |
| **Total deposited area (Mha)** | 1250.1 | 3070.7 | 3194.2 | 3398.5 | 3295 |
| **Total amounts (Tg N yr$^{-1}$)** | 14.0 | 47.2 | 60.7 | 70.5 | 83.5 |


**Table 2.** Comparison of manure and fertilizer N application amounts between this study and published
datasets.

| | Bouwman et al., (2002)[α] | Stehfest & Bouwman, 2006[β] | Bouwman et al., 2013[γ] | Chang et al., 2016[α] | Liu et al., 2010[γ] | Lassaletta et al., 2014[γ] | This study[γ] |
|---|---|---|---|---|---|---|---|
| **Manure N application (Tg N yr$^{-1}$)** | 12.4 | 4.8 | 57.8 | 12.4 | ~2.8 | NA | 7.6 |
| **Applied area (Mha)** | 625 | NA | 3358[η] | 1231 | NA | NA | 798 |
| **N fertilizer application (Tg N yr$^{-1}$)** | 4.3 | 3.1 | NA | 3.1 | 12.9 | 6.5 | 6.2 |
| **Applied area (Mha)** | 103 | NA | NA | 39 | NA | NA | 798 |

[α] estimated in 1995.
[β] national-level fertilizer data for 1998. The total grassland area for N fertilizer and manure was 677 Mha.
[γ] estimated in 2000.
[η] the grassland area includes both mixed and patrol systems.


**Figure 1.** Diagram of the workflow for developing the database of global annual N fertilizer use
rate in pasture during the period 1961–2016.

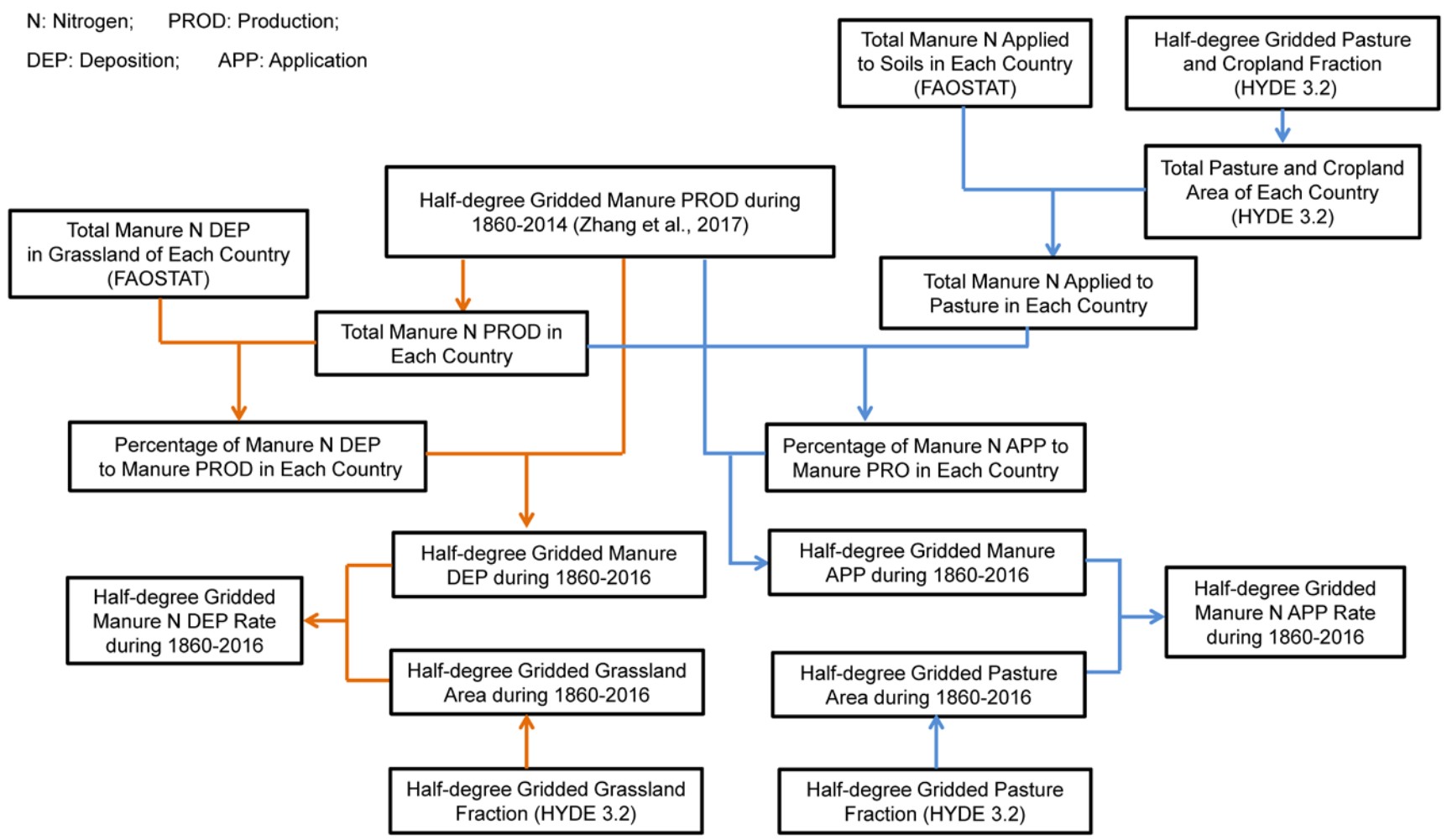


**Figure 2.** Diagram of the workflow for developing the database of global annual manure N use rate in pastures and manure N
deposition rate in pastures and rangelands during the period 1860–2016.


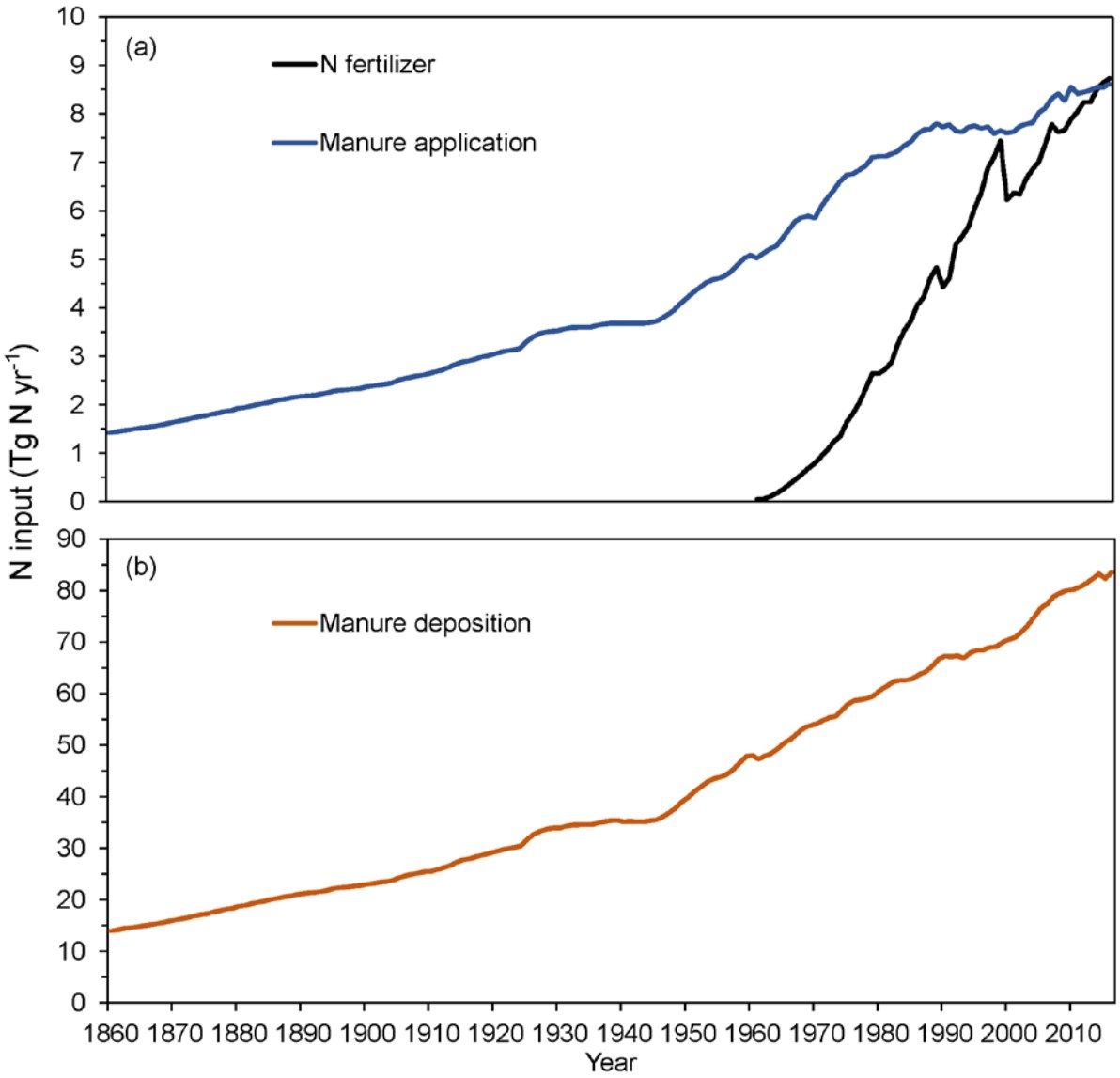


**Figure 3.** Temporal patterns of global manure N use, N fertilizer use, and manure deposition in
grassland systems: (a) Manure N use and N fertilizer use on global pastures during 1860−2016
and during 1961−2016, respectively; (b) Manure N deposition to global pastures and rangelands
during 1860−2016.

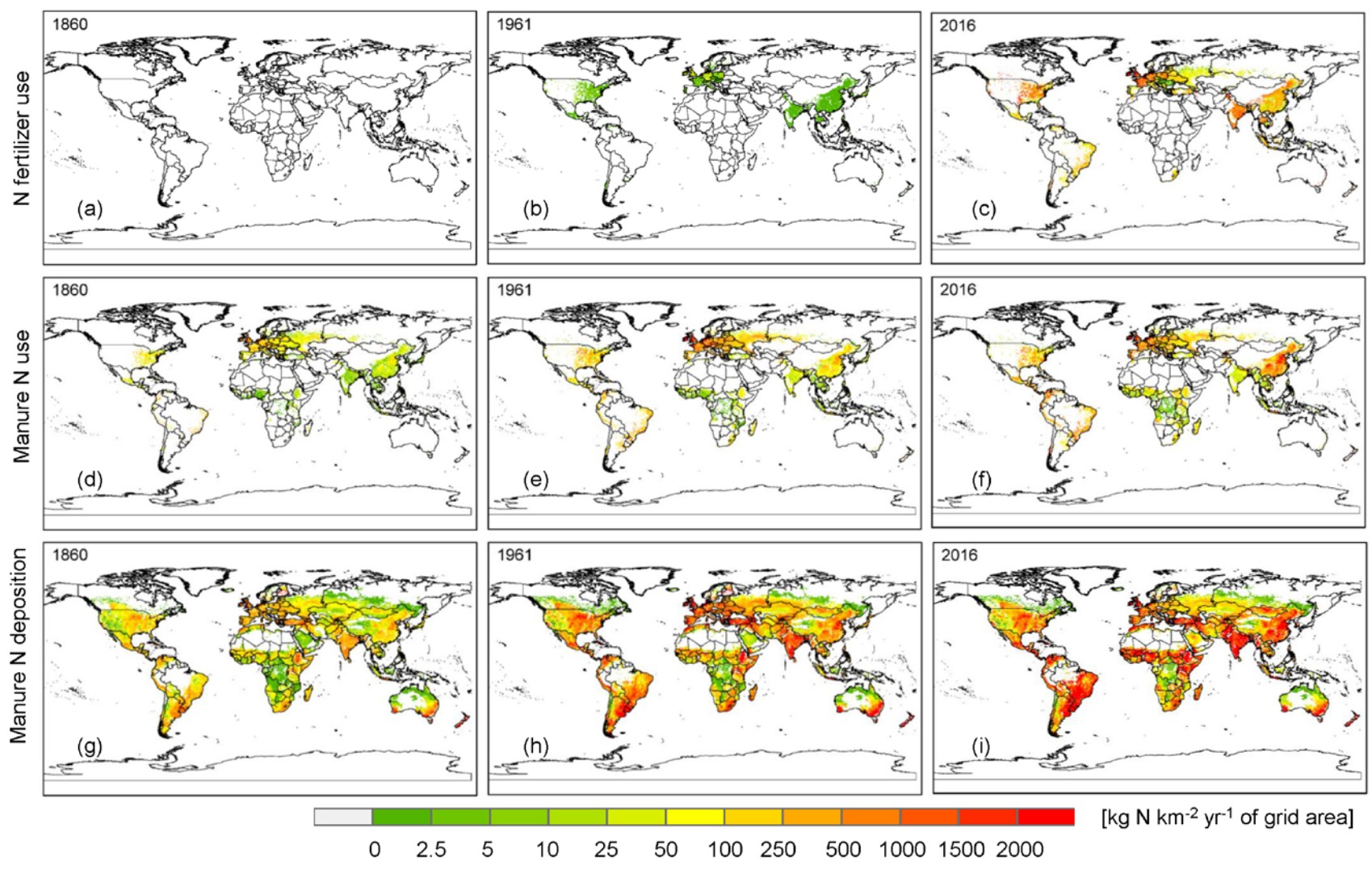


**Figure 4.** Spatial patterns of N input rates in global pastures and rangelands in 1860, 1961, and 2016: (a)–(c) N fertilizer application rates; (d)–(f) manure N application rates; (g)–(i) manure N deposition rates.

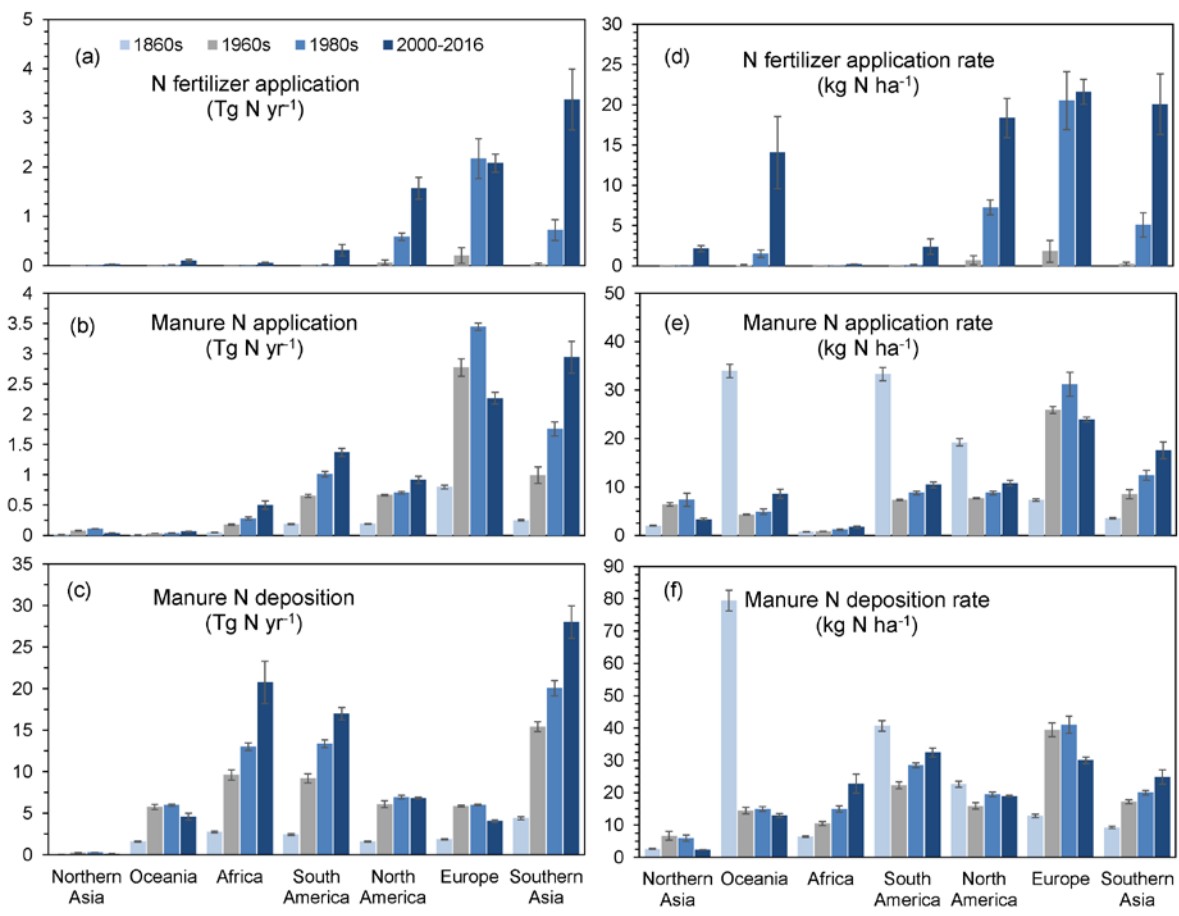

**Figure 5.** Nitrogen fertilizer use (a) and rate (d), manure N use (b) and rate (e), and manure N deposition (c) and rate (f) at regional scales in 1860s, 1960s, 1980s, and 2000–2016. Error bars represent standard deviation within each decade.