# Peer review of "Increased nitrogen enrichment and shifted patterns in the world's grassland: 1860-2016"

_Earth System Science Data, 2018_

## Referee Comment (RC1) · Anonymous Referee #1 · 26 Oct 2018

General comments:

The authors synthesized land-use and fertilizer-use datasets and produced spatially explicit maps of nitrogen (N) application across global grasslands. The study fills in the critical gap in global ecosystem and biogeochemical models to investigate N enrichment on grasslands. The analysis is well conducted, and the manuscript is well written.

I have two comments that may worth further exploring.

First, accounting for both the uncertainty and bias of the produced maps. Uncertainty refers to the confidence intervals around the point estimates. Note that none of this study's numbers come with uncertainty estimates. Uncertainty could come from the

source data, for example, fertilizer use in the early days (1860) is presumably less precise than the one in the recent days (2014). Uncertainty could also come from the process (see the next comment). In addition, bias refers to over- or under-estimate, for example, L404-405 mentions lack of human-induced sources N input will underestimate N input.

Second, the "spatialization" needs to consider the modifiable areal unit problem (MAUP) and spatial correlation. MAUP is a well-known issue in geography and spatial statistics, referring to the fact that when areal/point-based measures of spatial phenomena are aggregated into other units, they are influenced by both the shape and scale of the aggregation. It appears that the study simply takes the face values to fall into different spatial units without considering possible uncertainty and bias that are propagated through this process (coupled with the first comment). Also, note that the map estimates should somewhat consider spatial correlation. Spatial points are not independent: locations closer together will be more correlated than locations further apart.

Regardless of these improvements to be made, this study is an essential step in providing the global data and enhancing our understanding of the N issue.

Specific comments:

L155: the total number of countries is 202?

L578, Fig 4: Where are the boundaries of grasslands? The 3rd row appears to suggest grasslands cover the entire global landmass.
* * *

---

## Referee Comment (RC2) · Anonymous Referee #2 · 30 Oct 2018

Dear editors, dear authors,

Xu et al. presented a very important contribution fully in the scope of ESSD. The spatialized allocation of manure and synthetic fertilizers has been the Holy Grail for the researchers budgeting nutrients in agroecosystems of the world. The dataset presented here that is ready to be used open access represents a relevant step. I agree with the authors and there is still much work to do to improve accuracy and to solve many difficulties but this dataset is itself very valuable. I therefore recommend the publication of this manuscript-dataset in ESSD.

I have, anyhow, some generally minor comments that, in my opinion, should be considered by the authors before the work is ready for publication.

[Figure]

To strength the message I miss some extra figures:

1) To complement Figure 3 it would be nice another figure with the evolution of total surface of grasslands and also split into pasturelands and rangelands

2) Figure 5 should be completed with an equivalent extra figure (3 panels) including the inputs per ha in order to see the evolution of the intensification.

An uncertainty that for me is very important and it is not clearly highlighted is that inside a grassland cell we have pasture and rangelands with different livestock density and for sure different deposition rates. The final manure N deposition would be highly affected by the proportion of each type of management in the cell. Disentangling this point is relevant and, in my opinion, it is an important task for further research. Authors could remark this need for the future.

Other comments:

In the abstract please clarify "manure deposition by grazing animals", I am very used to prepare N soil budgets were the term "deposition" is used for atmospheric deposition and this clarification will improve the reading at a first sight.

Please, along the paper when you cite several papers, the order should be chronological (in the references section alphabetical)

L38 not only air but also nitrate leaching to water bodies

L48 not only meat but also dairy products (e.g. Bai et al. 2018 Global Change Biol)

L67 I am conscious that this dataset was probably developed for GHGs estimation but it will be useful for a wider audience (nutrient budgets in agricultural systems including ammonia emissions, leaching. . .)

L93 In supplements please include a table including the countries per region

L138-140 To help the reader please explain briefly the FAOSTAT methodology to estimate "manure applied to soils". Is NH3 emission discounted? Is manure dumped into the rivers discounted? (e.g. China, see Gu et al. 2015 PNAS).

In the results section please maintain the same order as in methods (i.e. manure deposition before application, or the other way around but consistently).

For the 3 inputs you provide the 5 top countries in terms of total input, I recommend to do the same with the input/ha to detect countries with a generalized high level of intensification.

L281-282 Please be careful when saying "total reactive N production of 217 TgN yr-1"), important part of the manure production has an origin in the synthetic fertilizer applied to feed crops or pasturelands, therefore is the same N recirculated into the system. You could say total "resulting in a total input of 217 TgN yr-1 and considering that it is in part recirculated" (or something similar with that message).

---

## Author Comment (AC1) · 10 Jan 2019

Dear reviewer #1, We thanks for the precious comments and constructive suggestions. These comments were addressed in detail and incorporated into the revised manuscript and supplementary material. All changes have been marked in "blue" to be tractable in the revised manuscript.

1. First, accounting for both the uncertainty and bias of the produced maps. Uncertainty refers to the confidence intervals around the point estimates. Note that none of this study's numbers come with uncertainty estimates. Uncertainty could come from the source data, for example, fertilizer use in the early days (1860) is presumably less precise than the one in the recent days (2014). Uncertainty could also come

from the process (see the next comment). In addition, bias refers to over- or under-estimate, for example, L404-405 mentions lack of human-induced sources N input will underestimate N input. Response: We totally agree with the reviewer that uncertainty should be considered in this study. In the main text, we are aware of your concerns and have discussed the uncertainty of this study from several aspects, as shown in section 4.5. We agree with the reviewer that fertilizer/manure use in the early days (1961/1860) is presumably less precise than the one in the recent days. However, this uncertainty is from the source data that beyond the scope of this study. It is hardly to provide uncertainty with numbers since our source data (i.e., FAOSTAT and Zhang et al. (2017)) have not provide uncertainty ranges. It is true that our datasets during 1961-2016 are more accurate as numbers are consistent with FAO national total N manure and fertilizer amounts. For the period 1860-1960, we assumed that the ratio of manure application/deposition to production is the same as for 1961. Combining with the gridded spatial maps of manure production rates in Zhang et al. (2017), we generated the datasets of spatialized manure application/deposition rates to global pastures/grasslands during 1860-1960. This has been described in sections 2.3 & 2.4. In addition, we have changed the section 4.5 "Uncertainties" to "Limitations and uncertainties" and will include uncertainty analysis in our future research.

2. Second, the "spatialization" needs to consider the modifiable areal unit problem (MAUP) and spatial correlation. MAUP is a well-known issue in geography and spatial statistics, referring to the fact that when areal/point-based measures of spatial phenomena are aggregated into other units, they are influenced by both the shape and scale of the aggregation. It appears that the study simply takes the face values to fall into different spatial units without considering possible uncertainty and bias that are propagated through this process (coupled with the first comment). Also, note that the map estimates should somewhat consider spatial correlation. Spatial points are not independent: locations closer together will be more correlated than locations further apart. Response: We thank for the reviewer's comments. The MAUP and spatial correlation mentioned above did not appear in this study. We aggregated the original maps of the

spatial resolution of 5 by 5 arc minute into 0.5 by 0.5 degree. The original spatial maps (HYDE 3.2 and Zhang et al. (2017)) are in raster formats and consist of pixels.

Regardless of these improvements to be made, this study is an essential step in providing the global data and enhancing our understanding of the N issue. Specific comments: L155: the total number of countries is 202? Response: The total number of countries with "manure N application" should be 165. We have updated this number in the main text. L578, Fig 4: Where are the boundaries of grasslands? The 3rd row appears to suggest grasslands cover the entire global landmass. Response: We have updated these figures. The boundaries are showing in Fig. 4g-i.

[Figure]

**Fig. 1.** Spatial patterns of N input rates in global pastures and rangelands in 1860, 1961, and 2016: (a)-(c) N fertilizer application rates; (d)-(f) manure N application rates; (g)-(i) manure N deposition rat

---

## Author Comment (AC2) · 10 Jan 2019

Dear reviewer #2,

We thanks for the precious comments and constructive suggestions. These comments were addressed in detail and incorporated into the revised manuscript and supplementary material. All changes have been marked in "blue" to be tractable in the revised manuscript.

1. To complement Figure 3 it would be nice another figure with the evolution of total surface of grasslands and also split into pasturelands and rangelands.

Response: Another figure with the evolution of total surface of grasslands, as well as the separated pastures and rangelands, has been added into the manuscript, as shown

in Fig. S2.

2. Figure 5 should be completed with an equivalent extra figure (3 panels) including the inputs per ha in order to see the evolution of the intensification. An uncertainty that for me is very important and it is not clearly highlighted is that inside a grassland cell we have pasture and rangelands with different livestock density and for sure different deposition rates. The final manure N deposition would be highly affected by the proportion of each type of management in the cell. Disentangling this point is relevant and, in my opinion, it is an important task for further research. Authors could remark this need for the future.

Response: We have added extra three figures including the inputs per ha into the manuscript, as shown in Figs. 5d-f. We also have added two tables (Tables S2 & S3) and another figure (Fig. S3) in Supplementary Material to show the changes of pasture and rangeland areas, and average N input rates in regional pastures and rangelands over the study period, respectively. We have also analyzed N input rates at regional scales and added them in section 3 of the main text. Line 239-243 "The average synthetic N application rate in Oceania, North America, and southern Asia showed a rapid increase over the period 1961-2016 (Fig. 5d). Africa and northern Asia showed a slight increase in average N fertilizer application rates during the study period. Europe exhibited a rapid increase of N fertilizer application rates since 1961, then decreased after 2000, and then started to increase in recent five years (Fig. S3)."

Line 272-277 "The regional average manure N application rate was increasing in southern Asia and Africa during 1860-2016 (Fig. S3b). South America, Oceania, and North America exhibited a rapid decreasing trend of manure N application rates from the 1860s to the 1960s and showed continuous increases afterward until 2016 (Figs. 5e, S3b), which was associated with the substantial expansion of pasture areas (Table S2). Europe exhibited a rapid increase of manure N application rates since the 1860s, then decreased after the 1980s (Figs. 5e)."

Line 306-313 "Oceania showed a continuously decreasing trend of average manure N deposition rates in pastures and rangelands over the period 1860-2016. Manure N deposition rates in South America decreased between 1860 and 1960 and then increased afterward until 2016 (Fig. S3c). The significant contrast of changes in manure N deposition rates in Oceania and South America between the 1860s and the 1960s is due to the substantial and rapid increase of grassland areas (Tables S2, S3). Africa and southern Asia saw continuous increases in manure N deposition rates from 1860 to 2016, whereas Europe and North America was found with decreasing deposition rates since the 1980s (Figs. 5f, S3c)." As shown in these figures, manure N application and deposition rates changed significantly associated with pasture and rangeland areas. Regions such as Oceania and South America experienced a substantial increase of pasture and rangeland areas, as described in Supplementary Material Text 2 (Table S2, S3). Due to this significant expansion, the manure N application and deposition rates are extremely high in the 1860s and decreased rapidly until the 1960s.

We totally agree with the reviewer and have already remarked your suggestions in section 4.5: "Last, inside each relevant land use cell pastures and rangelands may be characterized by different livestock density and deposition rates, which is not considered in our current datasets. The final manure N deposition would be highly affected by the proportion of each type of management in the cell. Thus, it is necessary to consider these in the future research."

Other comments: 1. In the abstract, please clarify "manure deposition by grazing animals", I am very used to prepare N soil budgets were the term "deposition" is used for atmospheric deposition and this clarification will improve the reading at a first sight.

Response: We have clarify it in the abstract: "We developed three global gridded datasets at a resolution of 0.5 degree by 0.5 degree for the period 1860-2016 (i.e., annual manure N deposition (by grazing animals) rate, synthetic N fertilizer use rate, and manure N application rate) by combining annual and 5-arc minute spatial data on pasture and rangeland with country-level statistics on livestock manure, mineral and

chemical fertilizers, and land use information for cropland and permanent meadows and pastures."

2. Please, along the paper when you cite several papers, the order should be chronological (in the references section alphabetical).

Response: We have checked the ESSD manuscript preparation guide, which shows "In terms of in-text citations, the order can be based on relevance, as well as chronological or alphabetical listing, depending on the author's preference." Thus, we still keep our citation orders in the main text.

3. L38 not only air but also nitrate leaching to water bodies.

Response: We have added it in the main text, as shown in lines 34-39, "There is a growing recognition that livestock production is linked to increasing global greenhouse gas (GHGs) and ammonia emissions (Tubiello et al., 2018; Xu et al., 2018). Unsustainable practices, especially in intensive systems, may lead to severe pollution of aquatic systems and soil degradation locally, regional and globally, in particular through nitrate leaching to water bodies (Dangal et al., 2017; Davis et al., 2015; Fowler et al., 2013; Yang et al., 2016)."

4. L48 not only meat but also dairy products (e.g. Bai et al. 2018 Global Change Biol).

Response: We have added it in the main text, as shown in lines 50-53, "Increased meat and dairy products consumption worldwide was a major driver behind the documented increase in cattle herds globally (FAOSTAT, 2018), and thus a major cause in the observed atmospheric increase of N2O and CH4 over the past several decades (Bai et al., 2018; Bouwman et al., 2013; Dangal et al., 2017; Tubiello, 2018)."

5. L67 I am conscious that this dataset was probably developed for GHGs estimation but it will be useful for a wider audience (nutrient budgets in agricultural systems including ammonia emissions, leaching. . .).

Response: We have added it in the main text, as shown in lines 81-83, "To enhance our

understanding of the role of grassland systems on the overall global GHG balance and nutrient budgets (e.g., ammonia emissions, nitrate leaching), global biogeochemistry models require spatially explicit estimates of N inputs."

6. L93 In supplements please include a table including the countries per region.

Response: We have added Table S1.

7. L138-140 To help the reader please explain briefly the FAOSTAT methodology to estimate "manure applied to soils". Is NH3 emission discounted? Is manure dumped into the rivers discounted? (e.g. China, see Gu et al. 2015 PNAS).

Response: We have added the brief description in the main text, as shown in lines 139-140, "Following IPCC guidelines, the data in this domain do not consider N leaching during treatment (FAOSTAT, 2018)."

8. In the results section please maintain the same order as in methods (i.e. manure deposition before application, or the other way around but consistently).

Response: We have changed the order in methods to maintain the same order as in results.

9. For the 3 inputs you provide the 5 top countries in terms of total input, I recommend to do the same with the input/ha to detect countries with a generalized high level of intensification.

Response: As we described above, we have analyzed N input rates at regional scales (Figs. 5d-f and S3a-c). As shown in these figures, manure N application and deposition rates changed significantly associated with pasture and rangeland areas (Tables S2 &S3). Due to this significant expansion, the manure N application and deposition rates are extremely high in the 1860s and decreased rapidly until the 1960s. The same changing patterns were found in some countries, especially within these two regions. We plot manure and synthetic N fertilizer input rates (kg N ha-1) for each country. Overall, an intensification of manure N application/deposition rates were seen in most

countries, however, the degree of intensification varied significantly. Manure N deposition rates exhibit a significant wide range, averagely from 0.2 kg N ha-1 in Iceland to 1178 kg N ha-1 in Bangladesh during 1961-2016. Similarly, manure N application rates averagely ranges from 0.3 kg N ha-1 in Congo to 160 kg N ha-1 in Netherlands during 1961-2016.

In terms of synthetic N fertilizer rates, although the values are the same within each country, they are also highly affected by each country's pasture areas. During 1961-2016, synthetic N fertilizer averagely ranges from less than 0.001 kg N ha-1 in several African countries (e.g., Congo, DRC) to higher than 500 kg N ha-1 in countries with extremely small amount of pasture areas but with relatively high amount of N fertilizer inputs (e.g., South Korea, Finland). Similarly, the rates are highly associated with country's pasture areas.

Manure N application/deposition and synthetic N fertilizer rates are within a large range and are highly associated with country's pasture/grassland areas. Moreover, the intensification of country average N application/deposition rates not only depends on the degree of increases in total N input amounts, but also the degree of changes (increase/decrease) in pasture/grassland areas. Thus, we thought it might be better to present country total amount of N inputs instead of input rates.

10. L281-282 Please be careful when saying "total reactive N production of 217 TgN yr-1"), important part of the manure production has an origin in the synthetic fertilizer applied to feed crops or pasturelands, therefore is the same N recirculated into the system. You could say total "resulting in a total input of 217 Tg N yr-1 and considering that it is in part recirculated" (or something similar with that message).

Response: Since we updated our datasets to 2016, numbers have been slightly changed. We have changed "total reactive N production of 217 TgN yr-1" into "resulting in a total input of 233 Tg N yr-1". Line 339-345 "During 2000-2016, the global mineral N fertilizer application to agriculture was significant, reaching 110 Tg N yr-1

in 2016, while manure N production was 123 Tg N yr-1 (FAO, 2018; FAOSTAT, 2018), resulting in a total input of 233 Tg N yr-1. Our estimate of total N inputs (synthetic N fertilizer: 7.5 Tg N yr-1; manure N application: 8.2 Tg N yr-1; manure N deposition: 78.1 Tg N yr-1) to permanent meadows and pastures (93.8 Tg N yr-1) accounted for 45% of global total N production (manure: 114.2 Tg N yr-1; synthetic N fertilizer: 96.4 Tg N yr-1) during 2000-2016."

Please also note the supplement to this comment:
https://www.earth-syst-sci-data-discuss.net/essd-2018-94/essd-2018-94-AC2-supplement.pdf
* * *
[Figure]

**Supplement:**

1. Regional classifications

According to the Intergovernmental panel on Climate Change the fifth assessment (IPCC AR5) and Tian et al. (2016), we divided the world into seven regions including North America, South America, Africa, Europe, southern Asia, northern Asia, and Oceania. To be clarified, southern Asia was divided into five parts (i.e., West, South, East, Central and Southeast Asia) since sub-regions within it have become hotspots for nitrogen inputs and greenhouse gas emissions (e.g., South and East Asia) (Fig. S1, Table S1).

Figure S1. The classification of regions in the world.

Table S1 The countries included in eleven regions across the globe.

| Regions | | Countries |
|---|---|---|
| North America | | Bahamas, Belize, Canada, Costa Rica, Cuba, Dominican Republic, EI Salvador, Guadeloupe, Guatemala, Haiti, Honduras, Jamaica, Martinique, Mexico, Nicaragua, Panama, Puerto Rico, Saint Lucia, Saint Vincent and the Grenadines, the US, and Trinidad and Tobago |
| South America | | Argentina, Bolivia, Brazil, Chile, Colombia, Ecuador, Guyana, French Guiana, Paraguay, Peru, Suriname, Uruguay  and Venezuela |
| Europe | | Albania, Austria, Belarus, Belgium, Bosnia and Herzegovina, Bulgaria, Croatia, Czech Republic, Denmark, Estonia, Finland, France, Germany, Greece, Hungary, Ireland, Italy, Latvia, Lithuania, Luxembourg, Macedonia, Malta, Moldova, Montenegro, Netherlands, Norway, Poland, Portugal, Romania, Slovakia, Slovenia, Spain, Sweden, Ukraine and the UK |
| Africa | | Algeria, Angola, Benin, Botswana, Burkina Faso, Burundi, Cabo Verde, Cameroon, Central African Republic, Chad, Democratic Republic of the Congo, Republic of the Congo, Cote d'Ivoire, Djibouti, Egypt, Equatorial Guinea, Eritrea, Eswatini, Ethiopia, Gabon, Ghana, Guinea, Guinea-Bissau, Kenya, Lesotho, Liberia, Libya, Madagascar, Malawi, Malvinas, Mali, Mauritania, Morocco, Mozambique, Namibia, Niger, Nigeria, Rwanda, Senegal, Sierra Leone, Somalia, South Africa, South Sudan, Sudan, Tanzania, Togo, Tunisia, Uganda, Zambia, and Zimbabwe |
| Oceania | | Australia and New Zealand |
| northern Asia | | Russian Federation |
| southern Asia | Central Asia | Kazakhstan, Kyrgyzstan, Tajikistan, Turkmenistan, and Uzbekistan |
| | East Asia | China, Japan, Mongolia, North Korea, and South Korea |
| | South Asia | Afghanistan, Bangladesh, Bhutan, India, Nepal, Pakistan, and Sri Lanka |
| | Southeast Asia | Cambodia, Indonesia, Laos, Malaysia, Myanmar, Papua New Guinea, Philippines, Thailand, Timor-Leste, Vietnam |
| | West Asia | Armenia, Azerbaijan, Georgia, Iran, Iraq, Israel, Jordan, Kuwait, Oman, Qatar, Saudi Arabia, Syria, Turkey, United Arab Emirates, and Yemen |

2. Global and regional grassland area changes over the period 1860-2016 (adapted from HYDE 3.2, Klein Goldewijk, 2017)

We aggregated 5-arc minute HYDE 3.2 land use dataset to 0.5 degree. Since HYDE 3.2 only provided every 10 years land use data during 1860-2000, we used the linear interpolation to produce annual maps for pastures and rangelands for each 0.5 grid cell. Grassland area increased from 1250 to 3295 Mha during 1860-2016, as shown in Fig.S2. The pasture area was increasing from 268 to 803 Mha during the study period (Fig.S2, Table S2). Compared with the area in the 1860s, Oceania and South America experienced a substantial expansion of pastureland, roughly 3136% and 2228%, respectively. North America, Africa, and southern Asia also exhibited a huge increase of pastureland, 760%, 381%, and 140%, respectively. In contrast, pastureland in Europe exhibited a slight decrease (13%). The rangeland area was increasing from 982 to 2492 Mha during 1860-2016 (Fig.S2, Table S3). Similar to the pastureland expansion, Oceania and South America experienced a substantial increase in rangeland area, 1656% and 521%, respectively, followed by North America (350%) and southern Asia (123%). In contrast, northern Asia exhibited a slight decrease, about 7%.

Figure S2. The temporal variations of global total grassland, pastureland, and rangeland areas during 1860-2016 (adapted from HYDE 3.2, Klein Goldewijk, 2017).

Table S2 Pasture area changes (Mha) during the period 1860-2016 (adapted from HYDE 3.2, Klein Goldewijk, 2017).

| Pasture (Mha) | North America | South America | Europe | Africa | Oceania | Southern Asia | Northern Asia | Total |
|---|---|---|---|---|---|---|---|---|
| 1860s | 9.9 | 5.6 | 109.3 | 60.9 | 0.2 | 69.8 | 7.7 | 263.4 |
| 1880s | 18.0 | 9.0 | 121.7 | 69.9 | 0.5 | 64.1 | 10.9 | 294.1 |
| 1900s | 38.4 | 18.0 | 134.2 | 79.2 | 1.1 | 70.9 | 16.2 | 358.0 |
| 1920s | 46.3 | 32.5 | 133.8 | 92.8 | 2.6 | 80.7 | 16.4 | 405.1 |
| 1940s | 66.5 | 54.3 | 133.2 | 123.9 | 4.4 | 90.9 | 24.1 | 497.3 |
| 1960s | 86.8 | 89.4 | 107.1 | 213.3 | 7.4 | 116.8 | 12.0 | 632.8 |
| 1980s | 83.2 | 118.7 | 105.0 | 258.4 | 8.1 | 149.8 | 13.0 | 736.2 |
| 2000-2016 | 85.3 | 131.2 | 94.6 | 293.1 | 7.6 | 167.9 | 11.6 | 791.3 |

Table S3 Rangeland area changes (Mha) during the period 1860-2016(adapted from HYDE 3.2, Klein Goldewijk, 2017).

| Rangeland (Mha) | North America | South America | Europe | Africa | Oceania | Southern Asia | Northern Asia | Total |
|---|---|---|---|---|---|---|---|---|
| 1860s | 61.0 | 54.1 | 31.8 | 372.0 | 20.1 | 404.2 | 33.1 | 976.3 |
| 1880s | 89.4 | 69.7 | 36.0 | 402.9 | 45.6 | 406.9 | 37.8 | 1088.3 |
| 1900s | 157.5 | 107.7 | 37.7 | 433.8 | 100.8 | 472.8 | 43.2 | 1353.5 |
| 1920s | 174.3 | 158.1 | 38.7 | 466.8 | 178.1 | 545.4 | 43.7 | 1605.1 |
| 1940s | 242.6 | 223.0 | 37.7 | 526.9 | 277.8 | 607.4 | 51.3 | 1966.7 |
| 1960s | 287.9 | 291.4 | 31.6 | 674.2 | 389.7 | 734.1 | 23.2 | 24312.1 |
| 1980s | 265.7 | 312.5 | 32.8 | 634.8 | 394.6 | 810.3 | 24.6 | 2475.3 |
| 2000-2016 | 274.8 | 335.7 | 36.4 | 598.0 | 353.4 | 903.3 | 30.8 | 2532.4 |

Figure S3. The temporal patterns of average N input rates (kg N ha$^{-1}$) in regional pastures and rangelands during 1860-2016.